# EXPLORING QUIC DYNAMICS: A LARGE-SCALE DATASET FOR ENCRYPTED TRAFFIC ANALYSIS

## ABSTRACT

QUIC, a new and increasingly used transport protocol, addresses and resolves the limitations of TCP by offering improved security, performance, and features such as stream multiplexing and connection migration. These features, however, also present challenges for network operators who need to monitor and analyze web traffic. In this paper, we introduce *VisQUIC*, a labeled dataset comprising over $100,000$ QUIC traces from more than $44,000$ websites (URLs), collected over a four-month period. These traces provide the foundation for generating more than seven million images, with configurable parameters of window length, pixel resolution, normalization, and labels. These images enable an observer looking at the interactions between a client and a server to analyze and gain insights about QUIC encrypted connections. To illustrate the dataset's potential, we offer a use-case example of an observer estimating the number of HTTP/3 responses/requests pairs in a given QUIC, which can reveal server behavior, client–server interactions, and the load imposed by an observed connection. We formulate the problem as a discrete regression problem, train a machine learning (ML) model for it, and then evaluate it using the proposed dataset on an example use case [1].

## 1    INTRODUCTION

The rapid adoption of Quick UDP Internet Connections (QUIC) (Gratzer et al., 2016) as a transport protocol offers significant enhancements over traditional TCP, including improved security, performance, and features such as stream multiplexing and connection migration. These advancements, however, also introduce challenges for network monitoring and analysis, particularly in the context of encrypted traffic. Traditional methods of traffic analysis are less effective with QUIC due to its encryption, necessitating innovative approaches to follow and managing network performance as well as its effects on latency, error rates, and congestion control. Consequently, the development of a comprehensive and diverse dataset composed of QUIC traffic to various web servers is essential for thorough research. This paper introduces a large-scale dataset for QUIC traffic, representing a major step forward in understanding QUIC dynamics, especially given the limitations of traditional traffic monitoring techniques in the face of QUIC's encryption.

This paper proposes a dataset that considers the case of an observer listening to the channel between the QUIC client and server. The observer sees data packets being sent in both directions. The proposed dataset contains more than $100,000$ QUIC traces collected from more than $44,000$ websites during a four-month period, from various vantage points, using a page request workload. The proposed dataset offers significant value in both the networking and ML domains.

To demonstrate the dataset's potential, we present a use case of estimating the number of HTTP/3 (Bishop, 2022) objects a QUIC connection carries. This information can be useful for various applications. The most important application is HTTP/3 load balancing. A load balancer can successfully balance the load it assigns to different machines if it is able to estimate the load imposed by each connection (Shahla et al., 2024). This is difficult with HTTP/3, because the load balancer does not know how many requests are sent by a client to the server on different QUIC streams [2].

---

[1] The dataset and the supplementary material can be provided upon request.

[2] This is also difficult with HTTP/2, because multiple requests can also be sent by an HTTP/2 client over one TCP connection. In this case, different streams are implemented by HTTP and not by QUIC. The same approach proposed here for HTTP/3 over QUIC is applicable for HTTP/2 over TCP.

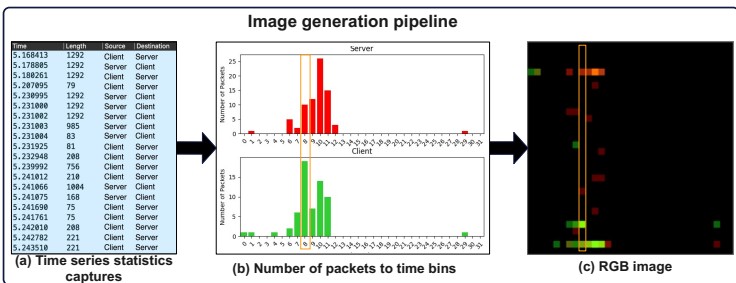

Figure 1: Generation of an image representing observed QUIC packets. The captured connection trace is windowed into overlapping temporal intervals. In each temporal window, the number of packets sent by the client and the server are binned into time bins. The obtained two-dimensional histograms (number of packets vs. time) are represented as an RGB image, with the green channel representing the packets sent by the client, the red channel representing the packets sent by the server, and the blue channel being unused.

The scheme presented in this paper can help to address this problem. Another use case is detecting HTTP/3 flood attack (Chatzoglou et al., 2023). In this attack, multiple HTTP/3 requests were sent to the server over a single connection. As indicated in Chatzoglou et al. (2023), identifying such an attack is challenging, because the attack pattern is almost identical to that of the normal traffic.

To address this aforementioned obstacle, this paper proposes a novel method for generating images from collected QUIC traffic traces, resulting in the *VisQUIC* dataset. By transforming QUIC data into a sequence of images, this approach enables ML models to analyze and predict network behaviors. The images are generated by "windowing" the connection traces into overlapping temporal intervals, binning the number of packets into time bins, and representing the resulting histograms as RGB images. The images' red and green channels indicate server-to-client and client-to-server packets, respectively, while the blue channel is unused. This technique enables the use of deep learning (DL) models to predict the number of HTTP/3 responses or requests in a given QUIC connection, as illustrated in Figure 1.

*VisQUIC* is created using more than $100,000$ QUIC traces collected from more than $44,000$ websites over a four-month period, resulting in a collection of over seven million images using two different window lengths. The length is a configurable parameter that can be fine-tuned when more images are added to the dataset. Having this dataset available facilitates increased comprehensive research on the behavior of HTTP/3 and QUIC, one of which, in the form of a use case, is presented in this work: estimating the number of HTTP/3 responses in the encrypted QUIC packets seen by an observer. The key contributions to our work are as follows:

- We release a dataset comprised of real-world $100,000$ traces from over $44,000$ websites page requests captured during a four-month period from various vantage points, using a page-request workload.

- We release and explain in detail how to generate learnable, customizable RGB images from real-world captured QUIC traces and create labels for them, resulting in over seven million labeled images.

- We demonstrate a glimpse of the potential use of the proposed dataset and provide a baseline algorithm to estimate the number of HTTP/3 responses in QUIC connections. The dataset can also be used for additional ML tasks.

The rest of this paper is organized as follows: Section 2 describes the dataset and image generation process. Section 3 defines the problem settings, and shows a baseline algorithm for estimating the number of responses in HTTP/3 traces using the proposed dataset. Section 4 reviews related work, and Section 5 concludes the paper.

## 2 DATASET DESCRIPTION

### 2.1 PROBLEM SETTINGS

We consider an observer who can see the QUIC encrypted packets transmitted from the client to the server and vice versa. For each packet, the observer knows its direction, length, and the observed time. With this information in hand, the QUIC traces can be converted into representative colored images, which are then suitable for training ML models. To convert the captured QUIC traces into time-series data, the sliding window technique Frank et al. (2001) is used. This technique requires two parameters: the window length and the overlap between consecutive windows. Both parameters are configurable.

### 2.2 TRACE COLLECTION AND IMAGE GENERATION

The process starts with HTTP/3 (Bishop, 2022) GET requests that are generated to various web servers that support HTTP/3, each hosting multiple websites. Requests are issued for up to 26,000 different websites per web server. Headless Chrome (chromium, 2017) is used in incognito mode with the application cache disabled, and the websites are requested sequentially. Table 1 displays the exact numbers for each web server and more detailed statistics broken down per web server for each class are provided in Appendix A.3. The generated network traffic traces are captured using Tshark (Merino, 2013) in packet capture (PCAP) format. These traces include only QUIC packets and cover the duration of the website request. For each PCAP file the corresponding SSL keys are stored to be used later to decrypt the traffic. The SSL keys are also provided in the dataset's materials.

Once we retain the time-series captured traces, the image datasets generation process can start for these traces. By Converting network traffic data—such as packet arrival times, packet sizes, packet density, and packet directions—into images the data is transformed into a format that is more compatible with DL models.

The use of images enhances pattern recognition abilities (Farrukh et al., 2023; Golubev & Novikova, 2022; Shapira & Shavitt, 2019; Tobiyama et al., 2016; Velan et al., 2015). Images enable the capture of complex interactions between features like packet sizes and arrival times within a two-dimensional space (server-to-client and client-to-server). This spatial representation allows DL models to identify intricate patterns that might be missed by traditional statistical or time-series analysis methods. For instance, correlations between packet bursts and response delays may become more discernible when visualized as variations in pixel intensity within an image.

Figure 1 shows an example of the construction steps for an image with a window length of $0.3$ seconds from a trace. During step (a), some of the trace statistics are collected: the time when the observer sees this packet, the packet's length, and the packet's direction. For a 0.3-second window, each bin contains 9.375 milliseconds. Step (b) shows histograms with $M = 32$ time bins for the considered window. The upper one is for the packets sent by the server and the lower one is for packets sent by the client. The horizontal axis represents the time bins and the vertical axis represents the number of packets received during each bin. For example, in the 8-th time bin (boxed in orange), the server sent 10 packets and the client sent 19 packets. Step (c) shows the image constructed for the considered example. The image represents the packet length statistics and the number of packets.

Figure 2 shows an example of the constructed image. The image is constructed on an $M \times N$ equispaced grid. The horizontal dimension represents different time window locations, while the vertical dimension represents different packet lengths. Thus, each packet is binned into one of the $M \times N$ bins according to its length and time. In the resulting image, the pixel at location $(i, j)$ represents the normalized number of packets whose length falls within the $j$-th bin received during the temporal span of the $i$-th time bin. The pixel's RGB values represent the normalized number of packets (i.e., density) sent from the server to the client (red) and from the client to the server (green). The blue channel is unused. The time interval spanned by the $i$-th bin is $[i\Delta t, (i + 1)\Delta t)$, where $\Delta t = T/M$ and $T$ denotes the window length. In our experiments, we used $T = 0.1$ and $T = 0.3$ seconds. To be counted in length bin $j$, the length of a packet should be in the range of $[j\Delta l, (j+1)\Delta l]$ with $\Delta l = L/N$ and $L = 1,500$ bytes denoting the maximum transmission unit (MTU). Histogram counts are normalized per channel window-wise using min-max normalization (Patro & Sahu, 2015), $x_{\mathrm{nrm}} = (x - x_{\min})/(x_{\max} - x_{\min})$, where $x$ and $x_{\mathrm{nrm}}$ are the original and normalized packet counts, respectively, and $x_{\min}$ and $x_{\max}$ are the minimum and maximum values of the packet count for the

specific direction in the considered window, respectively. The normalized value is multiplied by 255 to fit an 8-bit image format. If there is no traffic for a specific window, all pixels will contain the value zero. Note that the shortest QUIC packet is longer than what is represented by the first length bin. Therefore, the first row of the image grid consistently exhibits pixels with a value of zero.

Figure 2 shows different densities for each channel. For example, during time bin $i = 7$, different shades of green are displayed. This indicates that the client sent packets of five different lengths, which fall into bins $j = 2, 6, 12, 27$, and $28$. The five pixels are purely green, indicating that all the packets observed during bin $i = 7$ were sent by the client. The brightness of a pixel increases as its value approaches 255. Pixel $(7, 12)$ is the brightest across the whole window in the green channel and it represents 8 packets. This means that the largest number of packets sent by the client during the window is observed during time bin $i = 7$, when their length fell in bin $j = 12$. The other green pixels represent between 2 to 5 packets that are sent by the client. At time bin $i = 23$, the server sent packets of four different lengths, which are classified, based on their length, into bins $j = 7, 10, 15$, and $17$. The four pixels are purely red, indicating that during time bin $i = 23$, only packets sent by the server are observed. Pixel $(23, 10)$, representing 18 packets, is the brightest within the red channel across the entire window. The rest of the red pixels represent between 3 and 15 packets sent by the server. Pixel $(9, 26)$ is a combination of green and red, indicating that during time bin $i = 9$, packets from both the client and the server are observed and their length puts them into bin $j = 26$.

The image construction is an extension of the technique proposed by FlowPic (Shapira & Shavitt, 2019), which transforms network flows into images. For each flow, FlowPic creates an image from the packet lengths and the packet observed time. The goal of FlowPic is to construct a greyscale image using a flow-based two-dimensional histogram. FlowPic's single-channel approach, while providing a general traffic overview, is insufficient for more nuanced analysis, particularly in the context of QUIC. In QUIC, distinguishing between client-to-server and server-to-client traffic is critical due to the multiplexed nature of HTTP/3 requests and responses. Furthermore, QUIC's inherent complexity—stemming from stream multiplexing and independent packet handling—necessitates a more detailed examination of traffic directions. For those reasons we build upon the FlowPic technique and enhance it. We introduce a density factor for the packets' count in a given window and a configurable number of bins; in addition, we also separate channels, one for each direction. The result is an RGB image. Figure 4 shows different images using a different number of pixels. Using a higher number of pixels leads

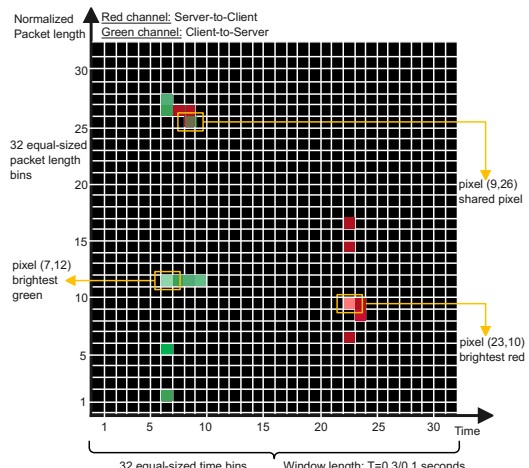

Figure 2: An image template, representing QUIC connection activity. Pixel positions represent histogram bins (horizontal and vertical axes corresponding to time and packet length, respectively). The values of the red and green channels represent normalized, per-window, histogram counts of the response and request packets, respectively.

to more detailed representation of the captured information. For example, a yellow pixel in Figure 4(a) which contains packets in both directions (a combination of the red and green channels), is split into more pixels as the resolution level is increased (Figures 4(b) and 4(c)), resulting in solely red or green pixels.

## 2.3 DATASET CREATION

For the creation of the image dataset in our example use case, several key parameters were defined: the sliding window length, normalization method, and pixel resolution. Specifically, we generated two image datasets with the following configurations: (1) two different sliding window lengths of $T = 0.1$ and $T = 0.3$ seconds; (2) images sized at $M = N = 32$, selected as a balance between resolution and computational cost. Using finer bins increased both the training and inference time with minimal accuracy improvement, while coarser bins negatively impacted model performance;

and (3) normalization applied per window rather than per trace. For the baseline example, we utilized a 90% overlap between consecutive windows during training, with no overlap during evaluation. The resulting labeled image dataset originates from over 100,000 traces collected from more than 44,000 websites, generating over seven million images.

**Labeling the images:** Each image in this dataset is labeled with the number of observed HTTP/3 responses; namely, the number of responses that have started to arrive within every time window. To this end, the SSL keys are used to decrypt the packets in a trace and reveal the packets' payloads. The HTTP/3 frames then are analyzed and HTTP/3 HEADERS frames are identified. Similarly, instead of labeling the images with the number of responses, the number of requests can be used as a label instead.

## 2.4 Training and test sets

The dataset can be split into two different settings: when the web servers are known to the observer and when they are not. In the former case, training and evaluation phases are done exclusively on the QUIC traces pertaining to the web servers assumed at inference time, using a $80 : 20$ ratio, ensuring out-of-training-sample evaluation. In the latter case, a leave-$x$-servers-out evaluation can be performed. For the provided use-case we show for the first setting results.

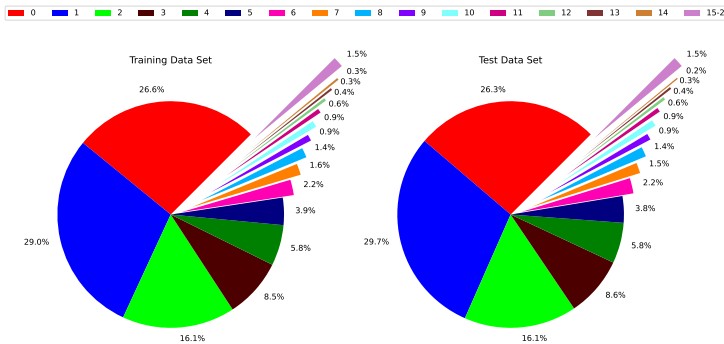

(a) $T = 0.1$ window dataset

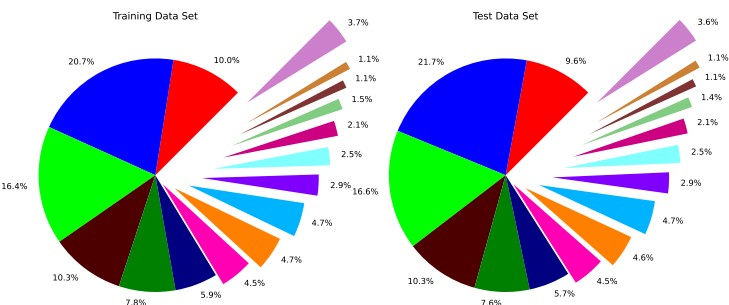

(b) $T = 0.3$ window dataset

Figure 3: Response distribution for training and evaluation datasets with two sliding window lengths ($T = 0.1$ and $T = 0.3$ seconds).

Figure 3 shows the images distribution for the created datasets. Both datasets are significantly skewed. As the figure demonstrates, images whose class values are $10$ or more are infrequent in both datasets. In the $T = 0.1$-second window dataset, labels $0, 1$ and $2$ make up roughly $75\%$ of the data, with the higher classes being represented in smaller proportions. Conversely, in the $T = 0.3$-second window dataset, there is a more even distribution, with labels $0, 1$ and $2$ comprising only about $47\%$ of the total dataset.

## 2.5 DISCUSSION:

**Selecting the window length:** The window length determines the temporal span each image represents, which directly impacts the data granularity. Shorter window lengths, such as 0.1 seconds, capture fine-grained temporal details of the network traffic, allowing for detailed analysis of short-term interactions between the client and server. This high granularity level is very useful for identifying subtle variations and transient behaviors in the traffic. Using shorter windows, however, also means generating a larger number of images per trace, leading to increased computational requirements. Conversely, longer window lengths, such as 0.3 seconds or more, offer a more aggregated view of the traffic, encapsulating longer sequences of packet interactions within each image. This approach reduces the number of images generated, thereby decreasing computational demands. Longer windows are beneficial for capturing broader trends and interactions over extended periods, which can be advantageous for understanding overall traffic patterns and behaviors. The trade-off between short and long window lengths is a potential loss of fine-grained details, which might be critical for certain types of predictions. The choice of window length, therefore, should balance the need for temporal resolution with the practical considerations of computational efficiency.

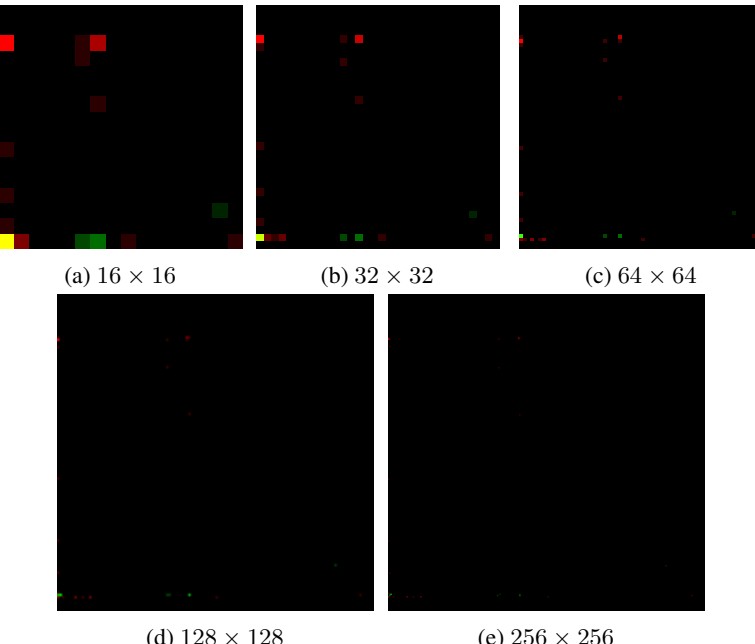

(a) $16 \times 16$       (b) $32 \times 32$       (c) $64 \times 64$

(d) $128 \times 128$       (e) $256 \times 256$

Figure 4: Five examples of the image representation of QUIC flows using $T = 0.1$-second windows and different pixels level.

**Choosing the Resolution Level:** Selecting the image size involves balancing resolution, computational efficiency, and the ability of ML models to extract meaningful features. Common image sizes, such as $32 \times 32$, $64 \times 64$, and $256 \times 256$ pixels, each offer specific trade-offs. A $32 \times 32$ image is highly efficient in terms of computation, storage, and processing speed, making it ideal for real-time analysis or situations with limited resources. However, the lower resolution may fail to capture complex network behaviors, which could limit model accuracy. Increasing the resolution to $64 \times 64$ pixels strikes a better balance between detail and computational efficiency. This resolution captures more intricate traffic features—such as packet inter-arrival times, traffic pattern variations, burstiness, and transmission rate changes—while keeping the processing overhead manageable. On the higher end, $256 \times 256$ images provide the finest level of detail, making them suitable for tasks requiring high precision and sensitivity to subtle variations in traffic. However, this higher resolution comes at the cost of increased computational demands, longer processing times, and greater storage requirements, which may be impractical for real-time or large-scale applications. Additionally, the interpretation of each pixel's resolution varies with window length, as each "bin" represents a portion of that window. Thus, the choice of resolution must be carefully aligned with the specific analysis goals and computational constraints.

**Normalizing per window versus per trace:** Normalizing the number of packet counts per window involves scaling the packet counts within each temporal window independently. This ensures that each window's data are scaled relative to its own range, which is beneficial for highlighting short-term variations and dynamics in network traffic. By normalizing per window, the resulting images maintain a consistent scale regardless of the overall trace length or variability across different windows or web servers. This approach helps mitigate the impact of outliers within individual windows, enabling the model to detect subtle differences in traffic behavior more effectively. However, normalizing per window can obscure broader trends and interactions that span across multiple windows, as each window is treated independently. In contrast, normalizing per trace involves scaling the packet counts across the entire trace before segmenting them into bins. This provides a uniform scale for all windows within a trace, preserving the relative differences across the entire connection. Normalizing per trace is advantageous for capturing long-term patterns and trends that persist throughout the trace. However, this may reduce sensitivity to short-term fluctuations, as the normalization is influenced by the extremes across the entire trace. Additionally, this form of normalization requires an offline analysis, making it unsuitable for online algorithms.

**Potential uses of the dataset:** The proposed dataset offers significant value not only in the networking domain but also in the ML field. From a ML perspective, it provides a novel way to represent complex, real-world phenomena—such as network traffic—through images. As outlined in this work, these images can be generated with varying resolutions, enabling researchers to study how different levels of granularity affect the performance of DL models, particularly those designed for image recognition. This aspect of the dataset opens up opportunities to investigate the optimal image resolution required for detecting intricate patterns in network traffic. For instance, researchers could generate an image from a trace using a set of parameters, $\mathbb{X}$, and compare it with another image from the same trace, using a different set of parameters, $\mathbb{Y}$ (e.g., resolution, density, normalization). Another key contribution is the structured nature of the dataset. Based on real-world QUIC traffic traces, it exhibits characteristics not commonly found in standard datasets, such as significant class imbalance, with certain labels appearing at very low frequencies (e.g., as low as 0.001%). Additionally, the dataset is well-suited for ordinal regression tasks, where the order of the labels is crucial. As demonstrated with our custom loss function (see A.1), predicting a label of 18 for a true value of 19 is closer to the correct answer than predicting 17 or 21, emphasizing the importance of maintaining label order. This unique structure makes the dataset valuable for exploring new methodologies and loss functions in ML research.

From a networking perspective, the contributions are even more direct. The dataset can be applied to a wide range of network-related analyses, from detecting DDoS attacks and traffic anomalies to examining symmetric and asymmetric flows, which could help identify the types of applications in use. It also holds promise for round-trip time (RTT) estimation and assisting load balancers in optimizing the distribution of network traffic, ultimately improving network management and performance.

## 3 ESTIMATING THE NUMBER OF HTTP/3 RESPONSES IN A QUIC CONNECTION

Estimating the number of HTTP/3 responses in a QUIC connection can assist a load balancer in making more informed decisions. By monitoring connections and estimating the number of responses within each connection, the load balancer can determine if a connection is considered heavy and adjust its decision on the selected server accordingly (Shahla et al., 2024).

To evaluate the use of the proposed dataset, we formulate the problem of estimating the number of responses in a QUIC connection as a discrete regression problem. It is not a classic classification task, because the misclassification errors depend on the distance between the categories. For example, consider an image with 17 responses. Estimating this number as 16 is better than estimating it as 15 or 19. It is also not a standard regression task, as the target categories are discrete. To address this issue, we developed a dedicated loss function coupled with data augmentation that considers: (1) the imbalanced dataset, which is derived from real-world QUIC traces and (2) rewarding the model for correctly predicting classes that are closer to the actual label than those that are farther away. Appendix A.1 explains the discrete regression loss function in more detail.

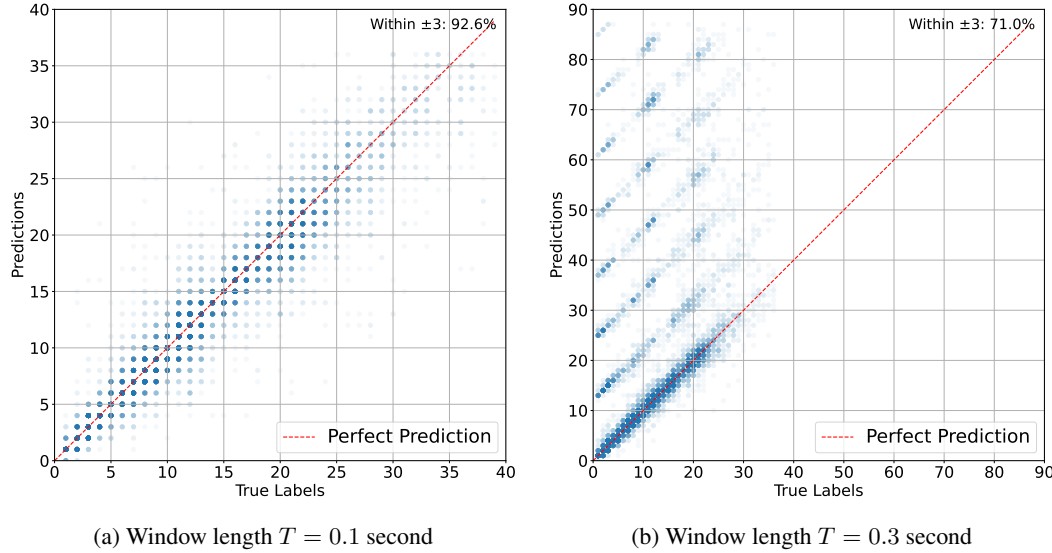

(a) Window length $T = 0.1$ second         (b) Window length $T = 0.3$ second

Figure 5: Scatter plots demonstrating the predictive results, where each point represents the summed predictions of a trace compared to its true label, with transparency set to $0.05$ to distinguish point density in overlapping areas.

We present a quantitative evaluation example of the proposed framework when the web servers are known to the observer, on a subset of the dataset, which was not present during training. A set of models were trained and evaluated exclusively on the QUIC traces pertaining to the web servers assumed at inference time. Two different models were trained with windows of $T = 0.1$ and $T = 0.3$ seconds. Classes with labels non-superior to 20 constitute $90\%$ of the traces in the $T = 0.3$-second window dataset and $95\%$ of the traces in the $T = 0.1$-second window dataset. Due to their scarceness, classes above 20 were excluded from the training and test sets.

To mitigate class imbalance, we developed a dedicated loss function and implemented a data augmentation technique. A grid search was performed to find the optimal values of $\alpha$, $\beta$, and $\gamma$ of the loss function . The values considered were $\alpha \in \{0.3, 0.5, 0.7\}$, $\beta \in \{0.4, 0.6\}$, and $\gamma \in \{1, 2, 3\}$. The optimal combination was chosen based on the lowest validation loss seen during the training process. The optimal values for $T = 0.3$ seconds were found to be $\alpha = 0.7$, $\beta = 0.4$, and $\gamma = 2$, while for $T = 0.1$ seconds, $\gamma = 3$ produced the best results with the same values of $\alpha$ and $\beta$. The training was performed with a batch size of $64$ images using the Adam optimizer (Kingma & Ba, 2014) with the ReduceLROnPlateau learning rate scheduler with a $30\%$ reduction in the learning rate, during the training phase. To reduce the risk of overfitting, an early stopping technique was used, with a patience parameter of six epochs. The performance is measured on an AMD Ryzen Threadripper PRO 3955WX 16C CPU 3.9G running at 64MB cache, 64GB of CRUCIAL CT8G4DFRA32A RAM clocked at 3200MHz and an NVIDIA GTX-4090 GPU.

The results presented are for estimating the total number of HTTP/3 responses in a complete trace. The images were fed sequentially through the trained models whose predictions were summed and compared to the sums of the trace's true label. Figure 5 shows prediction results on the using the evaluation traces, using the $T = 0.1$- and $T = 0.3$-second subdivisions. Both figures present a scatter plot in which the parameter $\theta$, ranging between 0 and 1, modulates the transparency of the plot. At $\theta = 0$, a point placed in the plot is fully transparent, whereas at $\theta = 1$, it is opaque. In these plots $\theta$ is set to $0.05$ to ensure high transparency and optimize the visual distinction between areas of high and low point density in cases of significant overlap among the roughly $12,000$ data points in each plot. In this plot, each point represents the summed labels or predictions over the images of a trace. For example, if a trace is composed of five non-overlapping images whose true labels are $1, 0, 2, 4$ and $1$, then the true label of that trace is $8$; if the model's predictions are $1, 0, 3, 4$ and $1$, for the same images, then the summed prediction is $9$, and that trace is represented in the plot as the $(8, 9)$ point, with $\theta = 0.05$ density. If another trace has the same aggregated values and is placed at the same $(8, 9)$ point, then it is placed on top of the previous point, thus making that point darker.

Additionally, we introduce a Cumulative Accuracy Profile (CAP) metric, which provides a refined measure of classification accuracy by incorporating a tolerance level for each prediction. Unlike traditional metrics such as confusion matrices that require exact matches between predicted and true labels, CAP allows for a specified degree of tolerance, accommodating predictions that are close to the correct class. Formally, it is defined as: $\text{CAP}_{\pm k}(\mathbf{y}, \hat{\mathbf{y}}) = \frac{1}{n} \sum_{i=1}^{n} \mathbb{1}(|y_i - \hat{y}_i| \leq k)$, where $\mathbf{y}$ represents the vector of true class labels, $\hat{\mathbf{y}}$ denotes the model's predictions, $k$ specifies the tolerance level (e.g., $\pm 1$ or $\pm 2$ classes), $n$ is the total number of samples, and $\mathbb{1}(\cdot)$ is the indicator function that evaluates to 1 if the condition is met and 0 otherwise. This metric thus quantifies the proportion of samples where the model's predictions fall within the allowed tolerance around the true class.

Figure 5(a) illustrates the scatter plot for predictions from the ML model trained using $T = 0.1$-second window images, while Figure 5(b) displays results for $T = 0.3$-second window. The test dataset includes $12,520$ traces with an average of $21.2$ images per trace for $T = 0.1$-second window images and $12,142$ traces with an average image of $7.5$ per trace for $T = 0.3$ seconds. The figures highlight significant improvements in the performance of the two ML models: first, the $T = 0.3$-second ML model has $71\%$ of predictions within $\pm 3$ (CAP) of a perfect prediction, whereas the $T = 0.1$-second model achieves $92.6\%$, demonstrating a nearly $20\%$ improvement in accuracy across entire traces. We use a $\pm 3$ tolerance level because for both window lengths, the points represent the aggregated prediction sum and, thus, the aggregated errors as well. The average number of images per trace is $7.5$ and $21.2$ for the $T = 0.3$-second and $T = 0.1$-second, respectively. Secondly, the predictions of the model that was trained using a $T = 0.1$-second window are notably more aligned along the diagonal, showing less deviation compared to those of the model that was trained using a $T = 0.3$-second window, suggesting that finer timing resolutions enhance the performance for the cumulative prediction.

Figures 5(a) and 5(b) illustrate a notable difference in predictive behavior between models that were trained and evaluated with $T = 0.3-$ and $T = 0.1-$second window sizes. Specifically, they show the presence of diagonal patterns in the predictions of the $T = 0.3$ model on the test set that are absent in the $T = 0.1$ predictions. This phenomenon exists for several reasons: (1) When using a $T = 0.1$ subdivision, a very high percentage of the images' true labels have lower class values, and the model that was trained using the $T = 0.1$ window dataset is very accurate for low value classes, whereas a $T = 0.3$ subdivision yields images with higher class values, hence increasing the variance of the true labels, when both models perform worse for the higher value classes as opposed to the lower class value; and (2) any incorrect prediction by either model contributes to an increase in the cumulative predictions for the remainder of the considered trace, thereby *elevating* the overall predicted values.

## 4 RELATED WORK

The study of QUIC and its impact on network traffic has gained significant traction in recent years due to its potential to enhance web performance and security. QUIC, developed by Google, aims to improve upon the limitations of traditional TCP by leveraging UDP for faster connection establishment and reduced latency (Almuhammadi et al., 2023). Numerous studies have explored various aspects of QUIC, including its interactions with encrypted DNS protocols such as DoT, DoH, and DoQ, and its integration with HTTP/3 (Zhou et al., 2022). These works have highlighted the performance benefits and challenges associated with adopting QUIC in diverse network environments. Other research has focused on the classification and analysis of QUIC traffic using advanced ML techniques. For example, ensemble learning models have been employed to classify QUIC network traffic with high accuracy, addressing the complexities introduced by QUIC's encryption features (Almuhammadi et al., 2023). Additionally, the implementation of QUIC in satellite communication has demonstrated its ability to improve performance metrics such as page load time and goodput, particularly when used in conjunction with performance enhancing proxies (Kosek et al., 2022).

CESNET-QUIC22 (Luxemburk et al., 2023) is a QUIC traffic dataset collected from backbone lines of a large internet service provider. It contains over $153$ million connections and $102$ service labels from one month of traffic. The dataset is fairly diverse, but lacks various important features. First, the metadata provided for packets such as direction, inter-packet time, and size is restricted solely to the first 30 packets of each connection, lacking comprehensive data for the entirety of the connections. Second, the lack of information regarding the HTTP/3 protocol renders it inadequate for tasks concerning studies focusing on the HTTP/3 protocol. In Smith (2021), a combined dataset of

TCP and QUIC traces is proposed. The traces were collected from three different VPN gateways worldwide. Our dataset contains traces collected from a single point, ensuring consistency in the dataset and allowing for a fair comparison between different traces and web servers. Additionally, we provide image representations for the dataset.

CAIDA (CAIDA, 2024) proposes a dataset composed of traffic trace collected from monitors on a commercial backbone link. However, the payload is removed from all packets, and only header information up to layer 4 (the transport layer) is kept. This again defies our purpose, as our dataset includes both the packet payload and the HTTP/3 protocol data.

## 5 CONCLUSION AND LIMITATIONS

In this paper, we introduced *VisQUIC*, a labeled dataset of QUIC traffic traces designed to facilitate advanced network behavior analysis, and ML tasks on real world data. By transforming QUIC connection data into sequences of RGB images, we leveraged DL models to effectively predict and analyze network traffic. We detailed the key decisions made during the dataset creation process, such as the selection of window length and image size, and emphasized the trade-offs between data granularity and computational efficiency. Our experimental results highlighted the effectiveness of the proposed approach, achieving accurate predictions of HTTP/3 responses within QUIC connections using image-based models. With normalized images per window, our models attained up to 97% CAP accuracy in scenarios where the web server was known. Additionally, we estimated the total number of HTTP/3 responses associated with each QUIC connection across more than 12,000 traces with a high accuracy of 92.6%. These findings demonstrated the power of image-based data representation for capturing complex network traffic patterns and improving network performance analysis. This method not only enhanced the ability to monitor and manage encrypted traffic but also paved the way for future research in network security and optimization. By offering a detailed and high-resolution perspective of QUIC traffic, the *VisQUIC* dataset served as a valuable resource for developing scalable and robust network analysis tools, driving innovation in the field.

**Limitations:** The dataset contains traces that are a result of web page requests done sequentially, one at a time. We use a page request workload because the number of web servers streaming video over QUIC is limited, leading to a dataset lacking diversity from the server perspective. We note that video streaming traffic patterns differ significantly from page requests, as they are heavily influenced by the streaming algorithms used by servers and not only the network conditions. Future work should study various bandwidths, using not only Chrome (Developers, 2023), but other browsers that support QUIC.

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

## A APPENDICES

### A.1 DEDICATED LOSS FUNCTION

For showing a use of our proposed dataset, we formulated the problem of estimating the number of HTTP/3 responses in QUIC connection as a discrete regression problem. The proposed loss function is,

$$L = \alpha \, \mathrm{FL} + (1 - \alpha) \left( (\beta \, \mathrm{ORL} + (1 - \beta) \mathrm{DBL} \right). \tag{1}$$

It comprises an aggregate of three terms: (1) a *focused loss* (FL) term, intended to alleviate class imbalance by minimizing the relative loss for well-classified cases while emphasizing difficult-to-classify ones; (2) a *distance-based loss* (DBL) term penalizing the model according to the predicted class's distance from the true label; and (3) an *ordinal regression loss* (ORL) term that introduces higher penalties for misclassifications that disrupt the natural ordinal sequence of the dataset, where lower class values occur more frequently.

The FL term (Lin et al., 2017) builds on the weighted cross-entropy loss (De Boer et al., 2005) by adding a focusing parameter, $\gamma$, which adjusts the influence of each sample on the training process based on the classification confidence. This parameter, $\gamma$, modifies the loss function by scaling the loss associated with each sample by $(1 - p_t)^\gamma$, where $p_t$ is the predicted probability of the true class $\mathbf{y}$. This scaling reduces the loss from easy examples (where $p_t$ is high), thereby increasing it for hard, misclassified examples, focusing training efforts on samples where improvement is most needed. Accordingly, the term is:

$$\mathrm{FL}(\mathbf{x}, \mathbf{y}) = \mathbb{E}_{(\mathbf{x},\mathbf{y})} \left[ -w(y) \cdot (1 - \hat{\mathbf{y}}_y(\mathbf{x}))^\gamma \cdot \mathbf{y}^\mathrm{T} \log \hat{\mathbf{y}}(\mathbf{x}) \right], \tag{2}$$

where $\mathbf{x}$ denotes the input sample, $\mathbf{y}$ is the one-hot encoded ground truth label, $\hat{\mathbf{y}}(\mathbf{x})$ represents the model's output of class probabilities, $\hat{\mathbf{y}}_y(\mathbf{x})$ denotes the predicted probability of the true class $y$, and $w(y)$ is a weight inversely proportional to the class frequency of $y$ in the training dataset. By assigning a higher weight to less frequent classes, the model places more emphasis on accurately classifying these classes during training. It is an effective strategy for dealing with class imbalance (Aurelio et al., 2019; Tian et al., 2020; Lin et al., 2017). FL thus minimizes the relative loss for well-classified examples, while emphasizing difficult-to-classify ones.

The DBL term (Wang et al., 2020)

$$\mathrm{DBL} = \mathbb{E}_{(\mathbf{x},y)} \left[ \sum_i \hat{y}_i(\mathbf{x}) \cdot |i - y| \right], \tag{3}$$

with $y$ denoting the ground truth class, is essentially a discrete regression loss that penalizes the model's output according to the predicted class's distance from the true label. The distance is computed as the absolute difference between the class indices and the target class.

Finally, the ORL term (Herbrich et al., 1999; Frank & Hall, 2001) is given by

$$\mathrm{ORL} = \mathbb{E}_{(\mathbf{x},\mathbf{y})} \left[ -\mathbf{y}^\mathrm{T} \log \sigma(\hat{\mathbf{y}}) - (1 - \mathbf{y})^\mathrm{T} \log \sigma(-\hat{\mathbf{y}}) \right], \tag{4}$$

with $\sigma$ denoting the sigmoid function saturating the input between $0$ and $1$. ORL uses a binary cross-entropy loss function, which compares the activation of each output neuron to a target that shows if the true class is greater than or equal to each class index, thus helping the model determine the order of the classes. Both DBL and ORL consider the relations between classes; they do so in different ways: DBL penalizes predictions based on the numerical distance, while ORL makes

Table 1: Summary statistics of QUIC traces and the number of images per dataset for each web server. Each web server containing multiple websites (URLs).

| Web Server | Websites | Traces | $T = 0.1$ | $T = 0.3$ |
|---|---|---|---|---|
| youtube.com | 399 | 2,109 | 139,889 | 54,659 |
| semrush.com | 1,785 | 9,489 | 474,716 | 221,477 |
| discord.com | 527 | 7,271 | 623,823 | 235,248 |
| instagram.com | 3 | 207 | 17,003 | 7,112 |
| mercedes-benz.com | 46 | 66 | 9,987 | 2,740 |
| bleacherreport.com | 1,798 | 8,497 | 781,915 | 331,530 |
| nicelocal.com | 1,744 | 1,666 | 148,254 | 48,900 |
| facebook.com | 13 | 672 | 25,919 | 10,988 |
| pcmag.com | 5,592 | 13,921 | 1,183,717 | 385,797 |
| logitech.com | 177 | 728 | 56,792 | 28,580 |
| google.com | 1,341 | 2,149 | 81,293 | 29,068 |
| cdnetworks.com | 902 | 2,275 | 207,604 | 85,707 |
| independent.co.uk | 3,340 | 3,453 | 176,768 | 68,480 |
| cloudflare.com | 26,738 | 44,700 | 1,347,766 | 341,488 |
| jetbrains.com | 35 | 1,096 | 34,934 | 18,470 |
| pinterest.com | 43 | 238 | 6,465 | 2,360 |
| wiggle.com | 4 | 0 | 0 | 0 |
| cnn.com | 27 | 2,127 | 91,321 | 59,671 |

explicit use of the classes' order. It focuses on preserving the correct order among predictions rather than the numerical distance between them.

The parameters $\alpha$, $\beta$, and $\gamma$ in the aggregated loss are used to balance the contributions of these three components to the combined loss. $\alpha$ is a parameter that controls the balance between the FL term and the ORL and DBL combination. A higher value of $\alpha$ gives more weight to the FL term, while a lower value gives more weight to the ORL and DBL combination. $\beta$ is a parameter that controls the balance between the ORL and DBL terms. A higher value of $\beta$ gives more weight to the ORL term, while a lower value gives more weight to the DBL term. $\gamma$ is a parameter used inside the FL component to adjust the focusing effect of the FL term. A higher $\gamma$ increases the effect of the focusing mechanism. This means the model pays more attention to correcting its worst mistakes, which is useful in highly imbalanced scenarios. Lower $\gamma$ values reduce the impact, making the loss more like a standard cross-entropy loss where each misclassification is weighted more uniformly.

### A.2 DATA AUGMENTATION:

Since the images are generated from QUIC traces that are formatted into a $32 \times 32$ pixel grid, each pixel corresponds to a unique feature of network traffic over a specific period. Any disruption in the temporal dependencies present in each image, such as through non-order-preserving modifications, may result in the loss of critical information, reducing the ML model's ability to estimate correctly. Thus, data augmentation is only applied to the minority classes (classes whose values are between 10 and 20), incorporating a minimal noise level (Maharana et al., 2022). We used noise with the standard deviation of $\sigma = 2.55$ corresponding to $1\%$ of the pixel value, ensuring that the added noise does not drastically alter the image appearance or disrupt the temporal dependencies. The noise serves, however, to imitate minor variations, increasing the model robustness and generalization capabilities.

### A.3 EXTENDED STATISTICS

### A.4 MOTIVATION

This dataset was created to study RTT estimation in the context of QUIC traffic over HTTP/3. To try to estimate RTT estimation, one needs to obtain information about response or request and to gather real QUIC traffic from various web servers, which currently is not researched a lot. QUIC is an encrypted protocol developed by Google, which is ran under Chrome browser.

Table 2: $T = 0.1$-second window dataset: Images per class value, per server.

| Class | youtube.com | semrush.com | discord.com | instagram.com | mercedes-benz.com | bleacherreport.com | nicelocal.com | facebook.com | pcmag.com | logitech.com | google.com | cdnetworks.com | independent.co.uk | cloudflare.com | jetbrains.com |
|---|---|---|---|---|---|---|---|---|---|---|---|---|---|---|---|
| 0 | 68,730 | 113,947 | 190,942 | 10,340 | 4,255 | 421,728 | 113,581 | 9,345 | 996,908 | 14,435 | 37,501 | 40,332 | 126,199 | 417,474 | 10,993 |
| 1 | 32,180 | 137,842 | 156,369 | 2,958 | 1,154 | 190,606 | 103,038 | 4,503 | 141,714 | 14,785 | 18,567 | 50,600 | 37,183 | 506,356 | 5,392 |
| 2 | 13,426 | 53,672 | 126,360 | 1,215 | 661 | 39,619 | 119,751 | 2,219 | 29,480 | 9,181 | 7,584 | 39,329 | 6,004 | 192,943 | 4,285 |
| 3 | 7,961 | 32,087 | 63,141 | 621 | 523 | 19,860 | 71,226 | 1,374 | 4,330 | 4,206 | 4,637 | 21,876 | 1,890 | 120,638 | 2,202 |
| 4 | 5,733 | 23,880 | 37,571 | 314 | 449 | 33,013 | 42,885 | 922 | 2,996 | 5,166 | 3,899 | 14,017 | 1,138 | 54,700 | 1,579 |
| 5 | 9,184 | 17,406 | 21,260 | 359 | 442 | 20,144 | 28,455 | 349 | 2,145 | 1,620 | 2,484 | 8,940 | 811 | 33,681 | 1,692 |
| 6 | 1,210 | 10,376 | 8,374 | 472 | 443 | 14,649 | 18,191 | 545 | 2,429 | 1,238 | 1,576 | 6,523 | 1,129 | 16,264 | 1,811 |
| 7 | 751 | 7,981 | 5,235 | 338 | 281 | 9,473 | 16,304 | 4,143 | 1,788 | 719 | 1,572 | 4,522 | 522 | 3,397 | 1,154 |
| 8 | 488 | 7,624 | 3,342 | 171 | 339 | 17,508 | 12,342 | 326 | 813 | 338 | 998 | 3,417 | 383 | 987 | 1,616 |
| 9 | 111 | 7,292 | 2,757 | 140 | 237 | 3,735 | 11,027 | 324 | 339 | 417 | 493 | 2,421 | 290 | 326 | 2,635 |
| 10 | 39 | 8,132 | 2,251 | 66 | 209 | 7,510 | 10,050 | 383 | 213 | 348 | 345 | 2,040 | 240 | 195 | 745 |
| 11 | 14 | 8,308 | 1,143 | 9 | 171 | 2,587 | 5,406 | 275 | 249 | 370 | 297 | 1,847 | 171 | 165 | 764 |
| 12 | 9 | 5,557 | 1,264 | 0 | 145 | 272 | 2,861 | 289 | 84 | 801 | 261 | 1,706 | 163 | 98 | 66 |
| 13 | 11 | 5,109 | 768 | 0 | 129 | 230 | 1,397 | 332 | 148 | 462 | 200 | 1,569 | 123 | 113 | 0 |
| 14 | 7 | 4,211 | 683 | 0 | 151 | 270 | 736 | 136 | 8 | 717 | 218 | 1,289 | 108 | 80 | 0 |
| 15 | 4 | 4,964 | 685 | 0 | 91 | 184 | 342 | 313 | 5 | 464 | 146 | 1,415 | 109 | 96 | 0 |
| 16 | 6 | 6,328 | 318 | 0 | 68 | 201 | 123 | 48 | 41 | 473 | 169 | 1,437 | 68 | 74 | 0 |
| 17 | 8 | 6,029 | 311 | 0 | 83 | 176 | 125 | 55 | 18 | 441 | 99 | 1,229 | 64 | 55 | 0 |
| 18 | 4 | 5,442 | 348 | 0 | 83 | 73 | 87 | 26 | 21 | 221 | 77 | 1,014 | 65 | 41 | 0 |
| 19 | 8 | 4,797 | 271 | 0 | 40 | 47 | 31 | 56 | 8 | 211 | 76 | 998 | 38 | 16 | 0 |
| 20 | 5 | 3,732 | 430 | 0 | 33 | 30 | 23 | 36 | 1 | 141 | 44 | 1,152 | 42 | 26 | 0 |
| Sum | 139,889 | 474,716 | 623,823 | 17,003 | 9,987 | 781,915 | 557,981 | 25,999 | 1,183,738 | 56,754 | 81,243 | 207,673 | 176,740 | 1,347,725 | 34,934 |

Table 3: $T = 0.3$-second window dataset: Images per class value, per server.

| Class | youtube.com | semrush.com | discord.com | instagram.com | mercedes-benz.com | bleacherreport.com | nicelocal.com | facebook.com | pcmag.com | logitech.com | google.com | cdnetworks.com | independent.co.uk | cloudflare.com | jetbrains.com |
|---|---|---|---|---|---|---|---|---|---|---|---|---|---|---|---|
| 0 | 11,185 | 23,751 | 26,130 | 2,942 | 1,082 | 107,871 | 19,851 | 1,537 | 316,252 | 3,990 | 7,031 | 8,530 | 31,494 | 45,521 | 3,100 |
| 1 | 12,954 | 56,849 | 22,566 | 1,081 | 214 | 70,024 | 21,724 | 1,510 | 35,476 | 5,041 | 6,702 | 13,318 | 26,392 | 86,815 | 1,713 |
| 2 | 6,569 | 34,403 | 31,911 | 655 | 168 | 37,090 | 23,963 | 640 | 21,239 | 4,459 | 4,465 | 11,463 | 4,968 | 84,493 | 1,622 |
| 3 | 4,651 | 18,272 | 33,139 | 359 | 105 | 18,401 | 23,611 | 917 | 5,272 | 2,663 | 1,954 | 7,993 | 2,075 | 37,014 | 1,005 |
| 4 | 3,447 | 13,116 | 33,214 | 286 | 78 | 11,685 | 18,049 | 768 | 1,427 | 2,261 | 1,617 | 7,114 | 928 | 22,280 | 722 |
| 5 | 6,307 | 12,567 | 23,249 | 143 | 71 | 7,682 | 15,305 | 312 | 768 | 1,629 | 912 | 6,084 | 491 | 14,270 | 732 |
| 6 | 3,745 | 7,098 | 17,279 | 237 | 88 | 6,711 | 12,300 | 167 | 1,077 | 2,182 | 860 | 5,844 | 292 | 13,965 | 741 |
| 7 | 2,918 | 5,823 | 18,919 | 541 | 67 | 8,433 | 9,786 | 2,254 | 1,114 | 739 | 1,293 | 5,140 | 363 | 12,478 | 532 |
| 8 | 1,439 | 4,921 | 13,591 | 238 | 76 | 22,045 | 8,065 | 193 | 1,047 | 511 | 979 | 4,414 | 246 | 10,979 | 1,025 |
| 9 | 815 | 3,624 | 4,322 | 189 | 71 | 12,548 | 7,170 | 63 | 468 | 477 | 529 | 3,350 | 153 | 7,975 | 1,642 |
| 10 | 230 | 3,917 | 2,848 | 190 | 78 | 13,965 | 7,394 | 69 | 358 | 342 | 370 | 2,717 | 117 | 2,784 | 1,505 |
| 11 | 166 | 5,221 | 1,651 | 167 | 62 | 10,530 | 7,527 | 61 | 551 | 335 | 396 | 2,024 | 173 | 954 | 2,383 |
| 12 | 80 | 4,704 | 1,261 | 77 | 50 | 2,464 | 8,338 | 114 | 267 | 360 | 373 | 1,754 | 103 | 641 | 1,677 |
| 13 | 46 | 3,430 | 882 | 7 | 50 | 478 | 9,361 | 185 | 103 | 289 | 317 | 1,336 | 125 | 443 | 71 |
| 14 | 24 | 2,194 | 988 | 0 | 66 | 248 | 10,034 | 713 | 169 | 982 | 274 | 1,055 | 110 | 290 | 0 |
| 15 | 29 | 2,725 | 775 | 0 | 71 | 341 | 7,660 | 293 | 75 | 549 | 273 | 909 | 133 | 152 | 0 |
| 16 | 14 | 3,827 | 669 | 0 | 62 | 232 | 5,217 | 192 | 25 | 444 | 188 | 752 | 58 | 125 | 0 |
| 17 | 10 | 4,288 | 431 | 0 | 60 | 369 | 3,471 | 185 | 33 | 516 | 169 | 626 | 51 | 88 | 0 |
| 18 | 10 | 3,641 | 479 | 0 | 75 | 170 | 2,207 | 356 | 50 | 387 | 152 | 469 | 71 | 93 | 0 |
| 19 | 11 | 3,300 | 408 | 0 | 77 | 104 | 1,177 | 424 | 14 | 262 | 132 | 375 | 78 | 80 | 0 |
| 20 | 9 | 3,806 | 536 | 0 | 69 | 139 | 759 | 35 | 12 | 162 | 82 | 440 | 59 | 48 | 0 |
| Sum | 54,659 | 221,477 | 235,248 | 7,112 | 2,740 | 331,530 | 222,969 | 10,988 | 385,797 | 28,580 | 29,068 | 85,707 | 68,480 | 341,488 | 18,470 |

## A.5 COMPOSITION

Each instance is an image associated with a label of the number of responses within that image. Each image in this dataset is labeled with the number of observed HTTP/3 responses; namely, the number of responses that have started to arrive within every time window. To this end, the HTTP/3 frames are analyzed and HTTP/3 HEADERS frames are identified. Similarly, instead of labeling the images with the number of responses, the number of requests can be used as a label instead.

Tables 1, 2 and 3 contain statistics for all images with a label of 20 or less. The files also contain images with high-class labels, and from the captured traces, additional images and labels can be generated. We publish the whole traces dataset, which contains $100,664$ traces in PCAP format, and outline the traces counts in Table 1. The traces were collected from various locations. The label for each image presented in the dataset is the number of responses observed during that time window of the image. Further studies can generate more images and more labels.

When using the traces or images in the dataset for training purposes, one should check if high-class labels of images are present on specific web servers before splitting the traces into sets of out-of-sample web servers. Some web servers lack images with high class values.

## A.6 COLLECTION PROCESS

The process starts with HTTP/3 (Bishop, 2022) GET requests that are generated to various web servers that support HTTP/3, each hosting multiple websites. Requests are issued for up to $3,176$ different websites per web server. Headless Chrome (chromium, 2017) is used in incognito mode with the application cache disabled, and the websites are requested sequentially. The generated network traffic traces are captured using Tshark (Merino, 2013) in packet capture (PCAP) format. These traces include only QUIC packets and cover the duration of the website request. Once we retain the time series captured traces, the image datasets generation process can start for these traces. The SSL keys are stored for each trace in a separate file (and are provided in the dataset). These keys are use to decrypt the relevant QUIC packets.

## A.7 PREPROCESSING/CLEANING/LABELING

Images whose class labels were above 20 responses were not part of the training or evaluation for estimating the number of responses in a QUIC connection, due to their rarity in the data. Besides that, the raw data contains all of the data, unfiltered. Preprocessing was done by filtering out packets that are not QUIC packets. Furthermore, a large number of images were identical, and all of the duplicate images were removed. Before the filtering there was $21,100,925$ images, and after there was $5,040,459$ images. We upload the full dataset including the duplicates. The raw data (captured traces) are saved and will also be available along with the images, upon the paper's acceptance using a link.

## A.8 OTHER USES

The dataset can be used for various communication tasks involving QUIC protocol, such as fingerprinting websites, estimating the number of requests in each connection, estimating the load on specific web servers, predicting server-client interactions over QUIC sessions, estimating RTT between a server and a client etc.

## A.9 MAINTENANCE

Both authors are maintaining the dataset on Github and on relevant links. The dataset can be updated and more labels can be added, for example, requests for each image in addition to responses. Moreover, the dataset can evolve to contain more images using different window lengths. Currently, the dataset contains two window lengths of 0.1 and 0.3 seconds. Errors may be submitted via the bugtracker on Github. More extensive augmentations may be accepted at the authors' discretion.

