# OpenReview forum: "Exploring QUIC Dynamics: A Large-Scale Dataset for Encrypted Traffic Analysis"
_ICLR.cc/2025/Conference — ICLR 2025 Conference Withdrawn Submission_

### Official Review · Reviewer_TCrS · 2024-10-29

**Soundness:** 1
**Presentation:** 2
**Contribution:** 1
**Rating:** 3
**Confidence:** 4

**Summary:**

This paper presents: 1. a labeled dataset of QUIC traces, 2. a method for converting those traces to 2D images, and 3. an application of the produced images to estimate the number of HTTP/3 objects that a QUIC connection carries.

1. The labeled dataset consists in 100k+ QUIC traces collected from 44k urls.

2. The conversion to images is done by looking, for each fixed interval of time, at the distribution of the packet length sent to the client (green channel) and sent to the server (red channel).

3. Finally, the authors propose an ordinal regression with a custom loss function for estimating the number of HTTP/3 objects that a QUIC connection carries.

**Strengths:**

- analyzing QUIC dynamics is an important problem
- 100k+ QUIC traces have been collected

**Weaknesses:**

- critical information is missing regarding the construction of the dataset (see questions below)
- the platform for collecting the dataset (=code) is not provided by the authors
- the dataset (which is the core of the paper) is not public but upon request
- the conversion of traces to images is not new
- regarding the regression problem, it is a classic regression problem, not an ordinal regression problem. The predicted variable is the number of observed responses, which is discrete but not relative. There is no comparison with alternative methods.
- in terms of writing style, a large part of the draft is too detailed: e.g. from line 166 to 207. lines 272 to 284 have been already discussed in line 204, etc.

**Questions:**

- Missing information regarding the construction of the dataset:
  + how many and where are the "various vantage points" (line 43)?
  + there are many possible configurations that have not been described: bandwidth, latency, packet loss, ...
  + how the 15 selected websites and the subset of urls have been motivated?
  + the data augmentation described in A.2 has never been mentioned in the paper

Additional comments:
- once printed, it is not possible to read the figures, e.g. Fig. 4
- the use of a pie chart in Figure 3 for representing an ECDF is incorrect
- the shape of Fig. 5 (b) and the explanation provided in line 454-462 is not convincing.

---

### Official Review · Reviewer_Ehsr · 2024-10-30

**Soundness:** 2
**Presentation:** 1
**Contribution:** 1
**Rating:** 3
**Confidence:** 5

**Summary:**

the paper presents a relatively small dataset,  tailored for the purpose of the counting objects in a QUIC stream over a network

the dataset consists of  an image-like representation of time evolving histograms of properties of the QUIC stream (packet size distribution at different time windows, for upstream and downstream direction), the labels is automatically extracted as the number of objects carried in the stream

the paper propose a baseline benchmark regression method based on a combination of three loss functions (to account for class imbalance, discreteness and ordering of the output)

**Strengths:**

the paper open source a dataset, which may have independent value  -- it is unclear though if the dataset scale and the choices made make this a relevant effort for the ICLR community ?

**Weaknesses:**

the paper has several limitations

- it is unclear if this type of paper is a good fit for the ICLR community: after all, ICLR stands for  international conference on learning representations, whereas here the representation is given ;  lexicographical jokes aside, it is unclear if this is an important dataset for the ICLR community, and to this reviewer opinion the answer leans on the negative side  (the labels is automatically extracted as the number of objects carried in the stream so the dataset is limited to "count quick object" tasks)

- the chosen representation  is, rather disturbingly, called an "image". while it is true that about anything can be represented as an image, the  representation is actually a time-evolving histograms of properties of the QUIC stream (different time windows on the x-axis, normalized packet size distribution on the y-axis, upstream and downstream direction on green/red channels; blue channel left unused). so it is fair to say that you could permute the order of the "rows" (i.e., bins of the normalized packet size distribution) and the information in the "image" would not be destroyed (while if you would do this for an actual image this would break spatial correlations).

- whereas the dataset fix the representation, it not even a "full image" as the blue channel is not used (nor the alpha channel), even though one could have easily imagined a use for it (e.g., the sum of traffic volume in the bin,  the distribution of inter-packet arrival, whatever other distribution). there are many hyper-parameters that have been fixed (such as the bin resolution ~ image width; the max-min scaling) which make the information extremely sparse (see Fig 4), so that one would have preferred a better approach, quantifying the loss of information from the raw data to this proposed representation,

- the dataset scale is relatively modest. as the process is automated (chrome browser downloading objects, traces captured and transformed in the above representation and object count extracted  automatically from decrypted traffic through browser stored keys), 100k sessions seems rather small. The existence of class imbalance does not come from "real world" traces, but also from the implicit choice of the experiment set (e.g., 2109 youtube traces vs 66 mercedes-benz.com vs 4 wiggle.com). The fact that the dataset is imbalanced forces authors to disregard traces with 20 objects (why? wasn't class imbalance a desirable feature before? was it not Focal Loss used to assess robustness against imbalance?)

- the dataset usefulness beyond the object count task is questionable. Authors mention the HTTP/3 flooding attack,  but is far less clear to which extent this dataset could be used as argued by authors: indeed, as we just noted, authors disregards connetions with as few as 20 objects, whereas it could be argued that the number of request per connection ina a HTTP/3 flooding attack would likely be significantly larger

- the benchmark model is a simple benchmark, so not fully developed, but is described nowhere and is even unclear how many trainable parameters are there in the model(s)?

**Questions:**

No further question beyond those listed in the weak points

**Details Of Ethics Concerns:**

No ethical concerns

---

### Official Review · Reviewer_P4W6 · 2024-10-30

**Soundness:** 2
**Presentation:** 3
**Contribution:** 2
**Rating:** 5
**Confidence:** 4

**Summary:**

The paper provides a labeled dataset for visualizing the Quic protocol.

The paper proposes a method to convert network traces into RGB images.

**Strengths:**

The paper provides a network dataset converted into images. The authors claim that the dataset can help understand the quiz protocol dynamics when the traffic is encrypted.

**Weaknesses:**

•	In the abstract, line 13, the authors state: _"These features, however, also present challenges for network operators who need to monitor and analyze web traffic."_ Yet, the paper does not clarify how the proposed method addresses this issue. Similarly, lines 33-35 mention, _"Traditional traffic analysis methods are less effective with QUIC due to its encryption, necessitating innovative approaches to manage network performance and its effects on latency, error rates, and congestion control."_ This issue, however, is expected for all encrypted network protocols using TLS. Please provide specific examples of how your proposed method addresses the challenges of monitoring encrypted QUIC traffic compared to conventional approaches.

•	The paper introduces a network dataset and its application in analyzing HTTP/3 traffic. Since ICLR is predominantly oriented towards advancements in machine learning, the paper may be better suited for a conference with a stronger emphasis on networking. The authors might consider submitting their work to a more networking-focused conference where the dataset's specific use case and its implications for network traffic analysis could be more thoroughly appreciated and critically assessed.

•	The "Labeling the Images" section (lines 221-225) of the paper presents a clear methodology for how the images are labeled with the number of HTTP/3 responses. However, the rationale behind needing a dataset labeled this manner remains unclear. It would be beneficial for the authors to explain why researchers might require a dataset that contains the number of responses per connection. Furthermore, the suggestion to alternatively label images with the number of requests raises an interesting point. Why not include both requests and responses within the same dataset? Integrating both data types could provide a more comprehensive view of the network dynamics, enabling easier identification of patterns such as DDoS attacks. Please consider discussing the feasibility and potential benefits of including both types of labels and any technical challenges or trade-offs involved in doing so.

•	The paper's primary contribution involves converting network traces into RGB images, a technique that, although presented as innovative, extends the already established FlowPic method. This approach has been in use for several years, reducing the novelty of the contribution. Moreover, while the QUIC network traces dataset is helpful, its exclusive focus on HTTP/3 limits its coverage of the diverse dynamics in QUIC traffic, such as video streaming or large file transfers. This reduces the dataset’s utility in various real-world scenarios where different data transmissions are critical. The authors should consider expanding the dataset to include other traffic types (e.g., video streaming and large file transfers). Additionally, the authors should discuss the challenges and potential benefits of expanding the dataset in this way.

•	The paper differentiates from the CESNET-QUIC22 dataset by including HTTP/3 protocol information. It's important to note that CESNET-QUIC22 encompasses a variety of labeled network traffic types. The authors propose that the conversion of CESNET-QUIC22 to images could be achieved using the same methodology. The authors should enhance the related work section to highlight the main differences between the works mentioned.

•	While the experiments effectively validate the parameters used in the proposed method for creating images, an analysis of the computational cost associated with image generation is absent. Understanding the computational demands is crucial, especially for real-time applications. The authors should consider including a detailed evaluation of the time and resources required to generate and process the images and discuss the implications for real-time network traffic monitoring and inference.

**Questions:**

***Minor comments***:

•	Line 131 By Converting --> By converting

•	Please increase the font size of Figure 3

---

### Official Review · Reviewer_pUWN · 2024-11-04

**Soundness:** 2
**Presentation:** 1
**Contribution:** 4
**Rating:** 3
**Confidence:** 4

**Summary:**

This paper introduces VisQUIC, a large-scale dataset comprising over 100,000 QUIC traces from more than 44,000 websites, collected over a four-month period. The dataset provides the foundation for generating over seven million images, with configurable parameters such as window length, pixel resolution, normalization, and labels. These images enable an observer to analyze and gain insights about QUIC encrypted connections, such as estimating the number of HTTP/3 response/request pairs in a given QUIC connection, which can reveal server behavior, client-server interactions, and the load imposed by an observed connection. The authors demonstrate a use case of estimating the number of HTTP/3 responses in the encrypted QUIC packets seen by an observer, formulating it as a discrete regression problem, and training a machine learning model using the proposed dataset.

**Strengths:**

* It Introduces a large-scale dataset comprising over 100,000 QUIC traces from more than 44,000 websites, collected over four months. Obviously, such work takes great effort, and it provides a valuable resource for the community to analyze QUIC traffic.

* It generates over seven million labeled images from the QUIC traces, with configurable parameters such as window length, pixel resolution, normalization, and labels. These images enable an observer to analyze and understand QUIC encrypted connections in detail.

**Weaknesses:**

* The presentation of this paper is feeble; in many cases, it looks more like a technical report than an academic paper. The followings illustrate some evidence:
  - A major issue is that the authors barely mention the motivation before conducting something. For a dataset paper, typically the first thing the authors should write about is "Why do we need a new dataset" (after the paper reviews the existing datasets): Does the network environment change a lot over the years? Are any emerging protocol features/traffic classes/attacks not included in other datasets? Do existing datasets miss fields that have become important nowadays? For the current version, readers have to read until page 9 to have a glimpse. I suggest the authors better clarify the motivation in the introduction, or add a section after reviewing existing datasets and before introducing your dataset.
  - Another example is the proposed image generation approach. The authors reminded the readers that this is not a novel method on page 3, and then started to introduce their image generation steps; the paper never deeply analyzes why we need to improve the existing generation approaches (until only two sentences on page 4, line 190).
  - Even if some issues regarding existing work are finally mentioned, these issues do not virtually motivate the design. For example, on page 10, line 490, the authors mention that CAIDA does not include packet payload. However, the images also cannot contain any payload, and the title says our topic is "encrypted traffic analysis" (which means the payload is assumed to be unavailable), so it feels unclear why to mention that.
  - In Figure 3, the authors never explain what is response distribution or the legend. The figures should be self-contained.
  - The paper does not distribute more space to the key contributions while writing too much detail (e.g., Section 2.5). It makes the readers hard to catch up with the focus of the paper.

* The evaluation only demonstrates that the new dataset works while does not show if it is better than the existing datasets in any aspect. For example, in the paper on the ToN_IoT dataset, the authors demonstrate a cross-training experiment fused with the Aposemat IoT-23 dataset to show better generalization that models can achieve by training on their dataset. Besides, more metrics in addition to accuracy are necessary, such as recall, precision, and F1 score.

* Anyway, I recommend the authors refer to the organization and presentation of the papers for those common datasets:
    - Toward Generating a New Intrusion Detection Dataset and Intrusion Traffic Characterization, 2018 (CIC-IDS 2017)
    - Towards the development of realistic botnet dataset in the Internet of Things for network forensic analytics: Bot-IoT dataset, 2019
    - ToN_IoT: The Role of Heterogeneity and the Need for Standardization of Features and Attack Types in IoT Network Intrusion Data Sets, 2022

* I believe the construction of such a large dataset is certainly meaningful to the community (that's why I give 4 to "contribution"). However, I may think the work is more meaningful to networking community than AI community (since it contains more protocol information rather than simplified features), so this submission might not be as a good fit as submitting to IMC/CoNEXT/INFOCOM.

**Questions:**

Please answer the aforementioned concerns.

---

### Note · Authors · 2024-11-21

I have read and agree with the venue's withdrawal policy on behalf of myself and my co-authors.